

# Hemiparasitic plants increase alpine plant richness and evenness but reduce arbuscular mycorrhizal fungal colonization in dominant plant species

Michael McKibben[1,2] and Jeremiah A. Henning[2,3,4]

[1] Department of Biology, University of Tampa, Tampa, FL, USA
[2] Rocky Mountain Biological Laboratory, Gothic, CO, USA
[3] Department of Ecology & Evolutionary Biology, University of Tennessee - Knoxville, Knoxville, TN, USA
[4] Department of Ecology, Evolution, and Behavior, University of Minnesota, Saint Paul, MN, USA

## ABSTRACT

Hemiparasitic plants increase plant biodiversity by reducing the abundance of dominant plant species, allowing for the establishment of subordinate species. Hemiparasites reduce host resources by directly removing nutrients from hosts, competing for light and space, and may indirectly reduce host resources by disrupting plant associations with symbiotic root fungi, like arbuscular mycorrhizal fungi and dark-septate endophytes. Here, we explored how a generalist hemiparasite, *Castilleja,* influences plant richness, evenness, community composition, and mycorrhizal colonization patterns across a ~1,000 m elevational gradient in the North American Rocky Mountains. We hypothesized that the presence of *Castilleja* would be associated with increased plant richness and evenness, shaping plant community composition, and would reduce mycorrhizal colonization within dominant plant taxa. However, the magnitude of the effects would be contingent upon climate contexts, that is, elevation. Overall, we found that the presence of *Castilleja* was associated with an 11% increase in plant richness and a 5% increase in plant evenness, regardless of elevation. However, we found that the presence of *Castilleja* influenced plant composition at only two of the five sites and at the remaining three of five sites, plot pairing was the only predictor that influenced composition. Additionally, we found that the presence of *Castilleja* reduced mycorrhizal fungal colonization within dominant plant species by ~20%, regardless of elevation. Taken together, our results suggest that hemiparasites regulate plant diversity, evenness, and interactions with mycorrhizal fungi independent of abiotic and biotic contexts occurring at the site, although overall effect on community composition is likely driven by site-level factors.

# INTRODUCTION

Root hemiparasitic plants regulate plant community composition and ecosystem function by reducing the abundance of dominant plant species, increasing co-existence of

Corresponding author
Jeremiah A. Henning,
jhenning@umn.edu

subordinate species, resulting in greater plant evenness (*Davies et al., 1997*; *Press & Phoenix, 2005*; *Bardgett et al., 2006*; *Westbury et al., 2006*; *Reed, 2012*). However, see *Press & Phoenix (2005)*, *Westbury & Dunnett (2007)* for examples that counter this general pattern. Hemiparasites range from host specialists to complete generalists, but typically impact dominant plant species to a greater degree compared to low abundant plant taxa (*Davies et al., 1997*; *Westbury et al., 2006*; *Reed, 2012*). However, generalist hemiparasites can have taxa specific effects independent of dominance patterns (*Adler, 2003*; *Demey et al., 2015*). Hemiparasites can directly reduce host plant biomass by parasitizing carbon and nutrients from the host through specialized structures called haustoria (*Adler, 2000*; *Salonen, Vestberg & Vauhkonen, 2001*). Although hemiparasites parasitize resources from their hosts, they are not wholly reliant on the hosts for carbon and nutrient resources and are able to fix carbon via photosynthesis and obtain soil resources through absorptive roots (*Ducharme & Ehleringer, 1996*; *Press & Phoenix, 2005*). As a result, hemiparasites impact hosts by directly parasitizing host resources in addition to competing for light and nutrients (*Press & Phoenix, 2005*). Additionally, hemiparasite associations with root-associated fungal communities like arbuscular mycorrhizal fungi (AMF) and dark septate endophytes (DSE) (*Davies & Graves, 1998*; *Li & Guan, 2008*) may provide an additional pathway for hemiparasites to acquire resources and compete with surrounding plant taxa.

Hemiparasite interactions with AMF and DSE may regulate or exacerbate the parasitic effects of hemiparasites on plant communities (*Gworgwor & Weber, 2003*; *Lendzemo et al., 2007*; *Bouwmeester et al., 2007*; *Li et al., 2012*). AMF and DSE are ubiquitous plant symbionts across most ecosystems globally and provide nutrient resources to plant hosts in return for plant-derived carbon, leading to the promotion of plant growth (*Smith & Read, 2008*; *Mandyam & Jumpponen, 2014*; *Vergara et al., 2017*; *Jumpponen et al., 2017*). Hemiparasites interact with AMF and DSE in a number of ways. For instance, hemiparasites may benefit from a host plant's association with AMF and DSE by indirectly accessing excess resources through the plant host, directly associating with the common mycorrhizal network, or by capitalizing on plant-mycorrhizal signaling pathways to colonize host roots (*Davies & Graves, 1998*; *Bouwmeester et al., 2007*; *Stein et al., 2009*; *De Vega et al., 2010*). If hemiparasites are superior competitors for host carbon resources, the reduction in available carbon resources may feedback to disrupt plant host and fungal symbionts interactions (*Stewart & Press, 1990*; *Davies & Graves, 1998*; *Press & Phoenix, 2005*). Conversely, plants may allocate more carbon resources to mycorrhizal fungi to increase nutrient access, offsetting hemiparasite effects. However, the effect of hemiparasites on AMF and DSE carbon allocation has not been directly tested. Conversely, AMF can inhibit the germination of hemiparasite seeds and the attachment of hemiparasites on host plants, reducing the effect of hemiparasites on host plants (*Gworgwor & Weber, 2003*; *Lendzemo et al., 2007*; *Li et al., 2012*). Thus, the interaction among hemiparasites, AMF, DSE, and plant hosts will likely have cascading effects on plant community diversity and composition, however, the impact of hemiparasites on AMF and DSE remains a significant knowledge gap.

Environmental contexts shape the composition and function of plant and fungal communities (*Callaway et al., 2002*; *Kivlin, Hawkes & Treseder, 2011*;

*Sundqvist, Sanders & Wardle, 2013*; *Jumpponen et al., 2017*). Because hemiparasites influence plant community richness, evenness, and community composition, it is likely that environmental contexts regulate the effect of hemiparasites on plant-fungal interactions (*Marx, Bryan & Davey, 1970*; *Press & Phoenix, 2005*; *Pennings & Callaway, 1996*, *2002*). For example, in low productive ecosystems, hemiparasites typically have a stronger influence on plant diversity and community composition because light competition with surrounding plants is low (*Matthies & Egli, 1999*; *Pennings & Callaway, 2002*; *Press & Phoenix, 2005*; *Těšitel et al., 2011*, *2018*). However, light competition in more productive ecosystems may outweigh negative parasitic effects, reducing the effect of hemiparasites (*Matthies & Egli, 1999*; *Pennings & Callaway, 2002*; *Press & Phoenix, 2005*; *Těšitel et al., 2011*, *2018*). Additionally, it has been hypothesized that hemiparasite effects on plant evenness and the promotion of subordinate species will be greatest in areas with high competitive asynchronies among taxa (*Pennings & Callaway, 1996*; *Press & Phoenix, 2005*). Thus, site-level abiotic and biotic factors likely regulate the effects of hemiparasites on plant community composition and associations with fungi.

Elevational gradients are ideal systems to understand how biotic and abiotic contexts shape species interactions, community composition, and ecosystem function (*MacArthur, 1972*; *Callaway et al., 2002*; *Sundqvist, Sanders & Wardle, 2013*; *Read, Henning & Sanders, 2017*). Moving from low to high elevation, abiotic properties like: temperature, precipitation, pH, nutrient availability, and biotic factors like: plant taxa present, herbivore abundance, symbiont abundance shift while often maintaining a similar disturbance history, soil parent material, and regional species pool (reviewed in *Körner, 2007*; *Sundqvist, Sanders & Wardle, 2013*). Together, biotic and abiotic factors shape the outcome of species interactions (*Callaway et al., 2002*) and productivity (*Whittaker et al., 1974*; *Sundqvist, Sanders & Wardle, 2013*) from low to high elevation. Across many elevational gradients, low elevation soil fungal communities are dominated by host carbon-reliant AMF communities, while high elevation ecosystems are often dominated by less host carbon-reliant DSE and ericoid mycorrhizal fungi (ERM) (*Haselwandter & Read, 1980*; *Schmidt et al., 2008*). Thus, the effect of hemiparasites on plant diversity, community composition, and soil fungal communities may shift across an elevational gradient, with effects on plant diversity and fungal colonization exacerbated at lower elevation communities (*Choler, Michalet & Callaway, 2001*; *Bardgett et al., 2006*; *Reed, 2012*).

In the North American Rocky Mountains hemiparasites in the genus *Castilleja* (Orobranchaceae) commonly occur in a wide-variety of montane ecotones (*Hersch & Roy, 2007*). *Castilleja,* in this region, are perennial, generalist hemiparasites (*Sweatt, 1997*; *Adler, 2003*), that receive up to 40% of their carbon from plant hosts (*Ducharme & Ehleringer, 1996*). *Castilleja* parasitize a wide variety of host plants, often increase plant community evenness, and increase nitrogen cycling in nutrient poor soil (*Ducharme & Ehleringer, 1996*; *Adler, 2003*; *Reed, 2012*; *Spasojevic & Suding, 2012*). Across an elevational gradient near Gothic, Colorado, USA, three species of *Castilleja*, *C. angustifolia* (2,480–2,740 m), *C. miniata* (3,392 m), and *C. sulphurea* (3,200–3,460 m) are distributed in different elevational clines. Here, we measured how the presence of

*Castilleja*: 1) was related to plant richness and evenness of the surrounding plant community, 2) would alter plant community composition, and 3) would reduce dominant plant associations with AMF and DSE at five sites along a ~1000 m pre-established elevational gradient (*Read, Henning & Sanders 2017*) near Gothic, Colorado, USA. We hypothesized that: (1) the presence of *Castilleja* would increase plant richness and evenness, (2) which would re-shape community composition and that (3) *Castilleja* would reduce the colonization of host carbon reliant AMF while increasing colonization of less carbon reliant DSE in the roots of the dominant plant species. However, (4) the effects of *Castilleja* on plant and fungal communities would be contingent on climate contexts, with stronger *Castilleja* effects at low elevation relative to higher elevation sites (increasing abiotic stress).

## METHODS

### Study site

We utilized pre-existing sites along an elevational gradient that spanned from 2,480 to 3,460 m (~1,000 m) near Gothic, CO, USA (*Read, Henning & Sanders, 2017*, Table 1). The sites are located on USDA Forest Service land and are covered under Forest Service Special Use Permit #GUN1120. The gradient site receives about 439–668 mm/year of precipitation and has a mean annual temperature of −1.6 to 1.5 °C (*Hijmans et al., 2005*; *Read, Henning & Sanders, 2017*; Table 1). Additionally, nutrient availability shifts across the gradient, with high elevation sites having more available phosphorus but lower available nitrogen relative to low elevation sites (*Read, Henning & Sanders, 2017*). Plant diversity is highest at middle elevation sites, while plant community composition transitions from a sagebrush steppe at low elevation to montane meadows at high elevation (*Read, Henning & Sanders, 2017*).

### Plant community sampling

At each elevation, we identified all *Castilleja* present within a 20 × 20 m area. Next, we haphazardly-selected 10 *Castilleja* individuals of similar size and identified an adjacent "*Castilleja* free" area that was located within two m from our focal *Castilleja*, but contained no *Castilleja* within a 1.5 m diameter around the centroid. Thus, we sampled 100 total plots (10 plots ×2 *Castilleja* treatments × 5 elevations). From our sampling design, we were unable to determine if plots were truly absent of *Castilleja* influence, as shallow-rooted *Castilleja* can infect across a semi-broad spatial area (*Ducharme & Ehleringer, 1996*). Logistical constraints of the pre-existing elevational gradient and the observational focus of our study prevented us from removing *Castilleja* individuals, which would have provided a more direct measurement of *Castilleja* effects on plant community composition and ecosystem function (*Reed, 2012*).

To measure plant community composition, we identified all plant species present within a 0.5 × 0.5 m quadrat in each plot. Plant community composition was measured in the *Castilleja* present plots, but placing the focal *Castilleja* in the center of our quadrat and *Castilleja*-free surveys were conducted by placing the quadrat in the center of the *Castilleja*-free area to maximize the distance from all surrounding *Castilleja* plants.

**Table 1 Site characteristics, dominant plant species, functional group of the dominant species, and root associated fungi within dominant plant roots.**

| Elevation (m) | 2,480 | 2,740 | 3,200 | 3,392 | 3,460 |
|---|---|---|---|---|---|
| Latitude | 38.65391 | 38.71533 | 38.96133 | 38.97005 | 38.99158 |
| Longitude | −106.86198 | −106.82264 | −107.03147 | −107.03987 | −107.06656 |
| MAT (C) | 1.36 | 1.52 | −0.80 | −0.70 | −1.62 |
| MAP (mm) | 443.4 | 439.2 | 599.0 | 592.0 | 667.8 |
| *Castilleja sp* present | *C. angustifolia* | *C. angustifolia* | *C. sulphurea* | *C. miniate C. sulphurea* | *C. sulphurea* |
| *Castilleja* cover (%) | 3.6 | 2.5 | 11 | *C. m.*—1.2 *C. s*—4.5 | 5.8 |
| Targeted species | *Balsamorhiza sagittata* | *Chrysothamnus viscidiflorus* | *Viola adunca* | *Ligusticum porteri* | *Arctostaphylos uva-ursi* |
| Functional group | Forb | Forb | Forb | Forb | Shrub |
| Fungi present | AMF, DSE | AMF, DSE | AMF, DSE | AMF, DSE | ERM, DSE |

Note:
MAT, mean annual temperature; MAP, mean annual precipitation; AMF, arbuscular mycorrhizal fungi; DSE, dark-septate endophytes; ERM, ericoid mycorrhizal fungi.

Next, we visually estimated the cover of each species present in the quadrat. We estimated percent coverage to the nearest 1% for species <20% and the nearest 5% for species >20% coverage. We also estimated the amount of bare ground and rock within each plot.

## Root sampling

From our cover data, we identified the most abundant plant species that occurred in all 20 plots at each elevation to sample for fungal colonization. At one of the five sites (2,480 m), our sampled plant species (*Balsamorhiza sagittata*) was also the dominant taxa, however, at four of the five sites, we sampled either the second or third most abundant taxa, as the dominant taxa did not occur in all 20 plots. We collected a single 2.5 × 15 cm core from our focal individual to measure the colonization of AMF and DSE. A single focal plant (Table 1) was haphazardly selected near the center of each plot (10 plots × 2 *Castilleja* treatments × 5 elevations = 100 total soil cores). We transported the soil cores back to the Rocky Mountain Biological Laboratory on ice and stored them at 4 °C in lab until being processed within 24 h. Next, we extracted live roots from the core using a 0.5 mm mesh sieve and placed roots in Fisher brand Histosette II tissue cassettes (Catalog #1000965, Thermo Fisher Scientific, Waltham, MA, USA). Tissue cassettes were then placed in deionized water to remove any remaining soil. Next, we cleared roots in a 10% potassium hydroxide solution, acidified root samples in a 2% hydrochloric acid solution, and then stained roots in a 0.01% trypan blue solution (*Koske & Gemma, 1989*). We then mounted roots on microscope slides using polyvinyl lactic acid glycerol glue (*International culture collection of (Vesicular) Arbuscular Mycorrhizal fungi (INVAM), 2017*) and slides were oven dried at 50 °C for 48 h. We removed two root samples (*Castilleja* present plant five and eight) from our 3,200 m elevation site because of low root biomass. Next, we quantified the presence of AMF hyphae, DSE hyphae, as well as AMF vesicles, arbuscules, coils, and spores, DSE microsclerotia, and any potential pathogens, of at least 50 root intersections per slide, using the magnified grid-line intercept

method (*McGonigle et al., 1990*). Finally, we calculated percent colonization of AMF, DSE, and any structures and potential pathogens by dividing the number of positive observations by the total number of root intersections observed.

## Statistical analysis

All analyses were conducted in R (*R Development Core Team, 2008*) and RStudio (*RStudio Team, 2015*), with packages cited within. To test for differences in plant richness, evenness, and community composition independent of the contribution of *Castilleja* we removed the *Castilleja* cover from all analyses. However, *Castilleja* cover is located in Table 1. We calculated species richness using the *specnumber* function in the *vegan* package (*Oksanen et al., 2017*) and species evenness as the probability of interspecific encounter (PIE, Simpson's Evenness) (*Hurlbert, 1971*) as: $PIE = N/(N - 1) \times (1 - \sum p_i^2)$, where $N$ = total number of individuals, and $p_i$ = is the relative abundance of species *i*. Within each plot using the *calcPIE* function in the *mobr* package (*McGlinn et al., 2018*). We chose to calculate PIE as our evenness measurement because of its independence from sample size when comparing across elevation sites (*Chase & Knight, 2013*). We tested for differences in plant richness and evenness between plots associated with the presence of *Castilleja* across sites by constructing linear mixed-effect models with elevational site, *Castilleja* presence (absent or present), and their interaction as fixed factors using the *nlme* package (*Pinheiro et al., 2014*). Within each mixed model, we allowed the intercept to vary by plot pairings (random effect). Next, we constructed mixed effect models with and without fixed factors (elevation and *Castilleja* presence) and compared AIC scores to determine if adding each fixed factor improved model fit. Next, we calculated the deviance and compared the inclusion of each factor with a likelihood ratio test using the *Anova* function (*car* package, *Fox & Weisberg, 2011*).

To test whether the presence of *Castilleja* was associated with changes in plant community composition, we performed a PERMANOVA using the *adonis* function in the *vegan* package (*Oksanen et al., 2017*) with *Castilleja* presence, elevation, and plot pairing as predictors. To separate the effects of *Castilleja* presence on abundance shifts among dominant taxa vs changes in taxa gains and losses, we performed PERMANOVAs using abundance-weighted Bray–Curtis distances as well as presence-absence data using Jaccard's distance. During our initial PERMANOVA fitting across all sites (Table S2), we observed significant interaction terms within both elevation × *Castilleja* presence ($F_{(1,99)} = 2.532$, $p = 0.01$) and elevation × plot pairing ($F_{(1,99)} = 13.48$, $p = 0.01$), which suggested that effect of *Castilleja* presence on plant community composition differed by site. To explore when and where *Castilleja* was associated with plant community composition, we conducted separate PERMANOVAs for each elevation. Within our site level PERMOANOVAs, we included *Castilleja* presence, plot pairing, and their interaction as predictor variables.

To visualize the results of our PERMANOVAs, we performed non-metric multidimensional scaling based on Bray–Curtis distances using the *metaMDS* function in the *vegan* package (*Oksanen et al., 2017*) for each elevational site.

**Table 2 Analysis of deviance table from "best-fit" mixed-model results exploring how site, the presence of *Castilleja*, and the interaction influence plant richness, plant evenness, and fungal colonization.**

| Response | Retained factors | $\chi^2$ | Df | p |
|---|---|---|---|---|
| Plant richness | ***Castilleja* present** | **52.360** | **1** | **<0.0001** |
| | **Elevation** | **38.905** | **1** | **<0.0001** |
| | *Castilleja* × Elevation | 0.4467 | 1 | 0.504 |
| Plant evenness (PIE) | ***Castilleja* present** | **6.8050** | **1** | **0.009** |
| | **Elevation** | **36.627** | **1** | **<0.0001** |
| | *Castilleja* × Elevation | 1.5041 | 1 | 0.220 |
| Mycorrhizal colonization (%) | ***Castilleja* present** | **163.17** | **1** | **<0.0001** |
| DSE colonization (%) | **Elevation** | **11.497** | **1** | **0.0007** |

Note:
Cast present, presence of *Castilleja*; Sum Sq, sum of squares; D*f*, degrees of freedom; *F*, *F*-statistic; *p*, *p*-value.
Bold factors represent statistically significant predictors for each response variable.

Finally, we compared fungal colonization patterns within the dominant plant species with mixed effect models as described above, with *Castilleja* presence/absence, elevational site and the interaction between *Castilleja* presence and elevation as predictor variables. For fungal mixed models, we constructed mycorrhizal (AMF + ERM) and DSE models separately. Structures of AMF and DSE (vesicles, arbuscules, coils, spores, microsclerotia) were extremely rare within our samples, therefore we present only AMF and DSE hyphal colonization data.

# RESULTS

## *Castilleja* association with plant richness and evenness

Overall, we found that the presence of *Castilleja* and elevation were retained within our best models to predict plant richness and evenness (Table S1). Taken as a whole, we found that the presence of *Castilleja* was associated with an increase in plant richness of 11% ($\chi^2$ = 52.360, $p < 0.0001$, Table 2), and an increase in plant evenness of 5% ($\chi^2$ = 6.805, $p = 0.009$, Table 2). As expected, elevation significantly impacted plant richness and evenness, with the highest plant species richness and evenness values observed at the middle elevation sites (3,200 m: richness μ = 11.8, evenness μ = 0.835; 3,392 m: richness μ = 11.2, evenness μ = 8.46) (Table 2; Fig. 1). Surprisingly, we observed no significant interaction terms between *Castilleja* presence and plant richness or evenness (Table 2).

## *Castilleja* association with plant community composition changes

Overall, we found that elevation was the best predictor of plant community composition change, accounting for 30% of the data variation in community composition (Table S2). Using abundance-weighted measures, *Castilleja* presence was associated with a significant ($p = 0.01$), but weak effect ($R^2 = 0.021$) on overall plant community composition (Table S2). Site-level heterogeneity (plot pairings) explained an additional 9% of data variation across plant community composition.

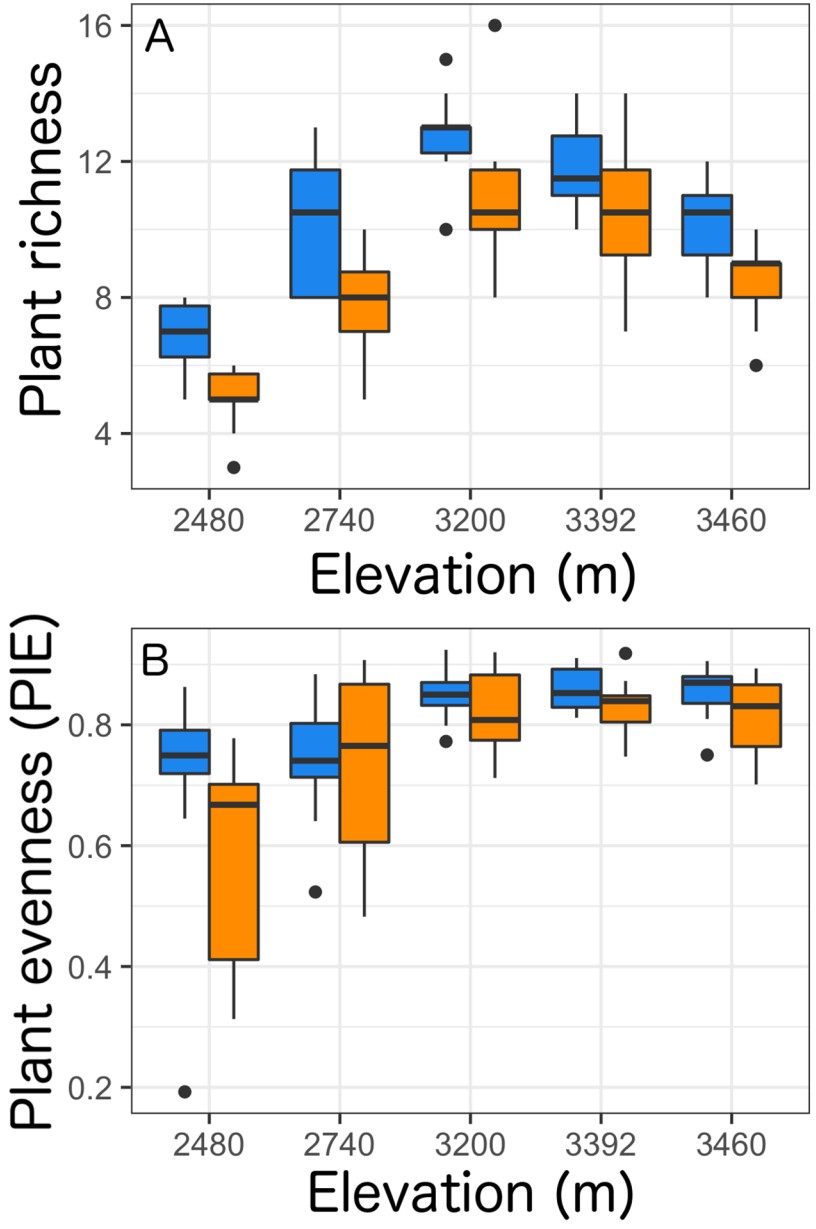

**Figure 1 Plant richness (A) and plant evenness (B, probability of interspecific encounter) within a 0.5 × 0.5 m quadrat across five elevational sites with *Castilleja* present (blue) or *Castilleja* absent (orange).** Each boxplot consists of ten plots at each elevation site. The midline represents the median value, with lower and upper hinges corresponding to the 25th and 75th percentiles. The upper and lower whiskers extend from the hinges to highest to the lowest values but no further than 1.5× the inter-quartile range (IQR).

At the site level, we found that the presence of *Castilleja* was associated with plant composition shifts in abundance-weighted Bray–Curtis distance at two (2,740 and 3,200 m) of the five elevation sites (Table 3; Fig. 2). At 2,740 m, we observed higher abundance of *Adenolinum lewisii*, *Phacelia sericea*, *Antennaria rosea* and reduced abundance of shrub species *Artemisia tridentate* and *Symphoricarpos rotundifolia*, and forb *Delphinium nuttallianum* when *Castilleja* was present (Fig. 2). At 3,200 m,

**Table 3 Site-level (elevation) abundance-weighted (Bray–Curtis) PERMANOVA results to partition the effect of *Castilleja* presence and spatial structure (plot pairing) on plant community composition.**

| Elevation | Factor | D*f* | SS | MeanSqs | *F* | *R*$^2$ | *p* |
|---|---|---|---|---|---|---|---|
| 2,480 m | *Castilleja* present | 1 | 0.211 | 0.21099 | 1.087 | 0.0529 | 0.37 |
| | **Plot pairing** | **1** | **0.500** | **0.50037** | **2.577** | **0.1255** | **0.05** |
| | *Cas.* present × Pairing | 1 | 0.170 | 0.16982 | 0.875 | 0.0426 | 0.54 |
| | Residuals | 16 | 3.106 | 0.19414 | | 0.7790 | |
| | Total | 19 | 3.987 | | | 1.0000 | |
| 2,740 m | ***Castilleja* present** | **1** | **0.291** | **0.29112** | **2.173** | **0.1004** | **0.03** |
| | Plot pairing | 1 | 0.243 | 0.24259 | 1.811 | 0.0837 | 0.14 |
| | *Cas.* present × Pairing | 1 | 0.222 | 0.22188 | 1.656 | 0.0765 | 0.09 |
| | Residuals | 16 | 2.143 | 0.13397 | | 0.7394 | |
| | Total | 19 | 2.899 | | | 1.0000 | |
| 3,200 m | ***Castilleja* present** | **1** | **0.481** | **0.48098** | **2.637** | **0.1257** | **0.01** |
| | Plot pairing | 1 | 0.278 | 0.27842 | 1.527 | 0.0727 | 0.11 |
| | *Cas.* present × Pairing | 1 | 0.150 | 0.14983 | 0.822 | 0.0391 | 0.55 |
| | Residuals | 16 | 2.918 | 0.18237 | | 0.7624 | |
| | Total | 19 | 3.827 | | | 1.0000 | |
| 3,392 m | *Castilleja* present | 1 | 0.101 | 0.10117 | 0.756 | 0.0355 | 0.69 |
| | **Plot pairing** | **1** | **0.523** | **0.52344** | **3.911** | **0.1837** | **0.01** |
| | *Cas.* present × Pairing | 1 | 0.084 | 0.08361 | 0.625 | 0.0293 | 0.76 |
| | Residuals | 16 | 2.141 | 0.13382 | | 0.7514 | |
| | Total | 19 | 2.849 | | | 1.0000 | |
| 3,460 m | *Castilleja* present | 1 | 0.229 | 0.22856 | 1.076 | 0.0536 | 0.37 |
| | **Plot pairing** | **1** | **0.546** | **0.54636** | **2.573** | **0.1280** | **0.04** |
| | *Cas.* present × Pairing | 1 | 0.094 | 0.09421 | 0.444 | 0.0221 | 0.90 |
| | Residuals | 16 | 3.398 | 0.21239 | | 0.7963 | |
| | Total | 19 | 4.267 | | | 1.0000 | |

**Note:**

D*f*, degrees of freedom; SS, sequential sums of squares; MeanSqs, mean squares; *F*, *F*-statistic; *R*$^2$, partial *R*$^2$; *p*, *p*-value. Bold factors represent statistically significant predictors for each response variable.

we observed reduced cover of invasive grass *Bromopsis inermis,* when *Castilleja* was present (Fig. 2). At three of the five sites (2,480, 3,392, and 3,460 m), we observed a significant relationship of plot pairing, suggesting that spatial heterogeneity in plant community composition may cloud our ability to observe the effect of *Castilleja* on community composition. Unlike our abundance-weighted results, we found no effect of *Castilleja* and only weak effects of plot pairings on plant community composition when sites were compared using presence-absence based Jaccard's index (Table S3).

## *Castilleja* association with mycorrhizal and dark-septate endophyte colonization

Overall, we found that *Castilleja* presence was associated with reduced mycorrhizal colonization at each site by 20% (Table 2; Fig. 3) and was the only factor retained within our best fit model (Table S1). At our four lowest elevation sites *Castilleja* presence was correlated with a reduction in AMF colonization in *Balsamorhiza sagittata* (2,480 m),

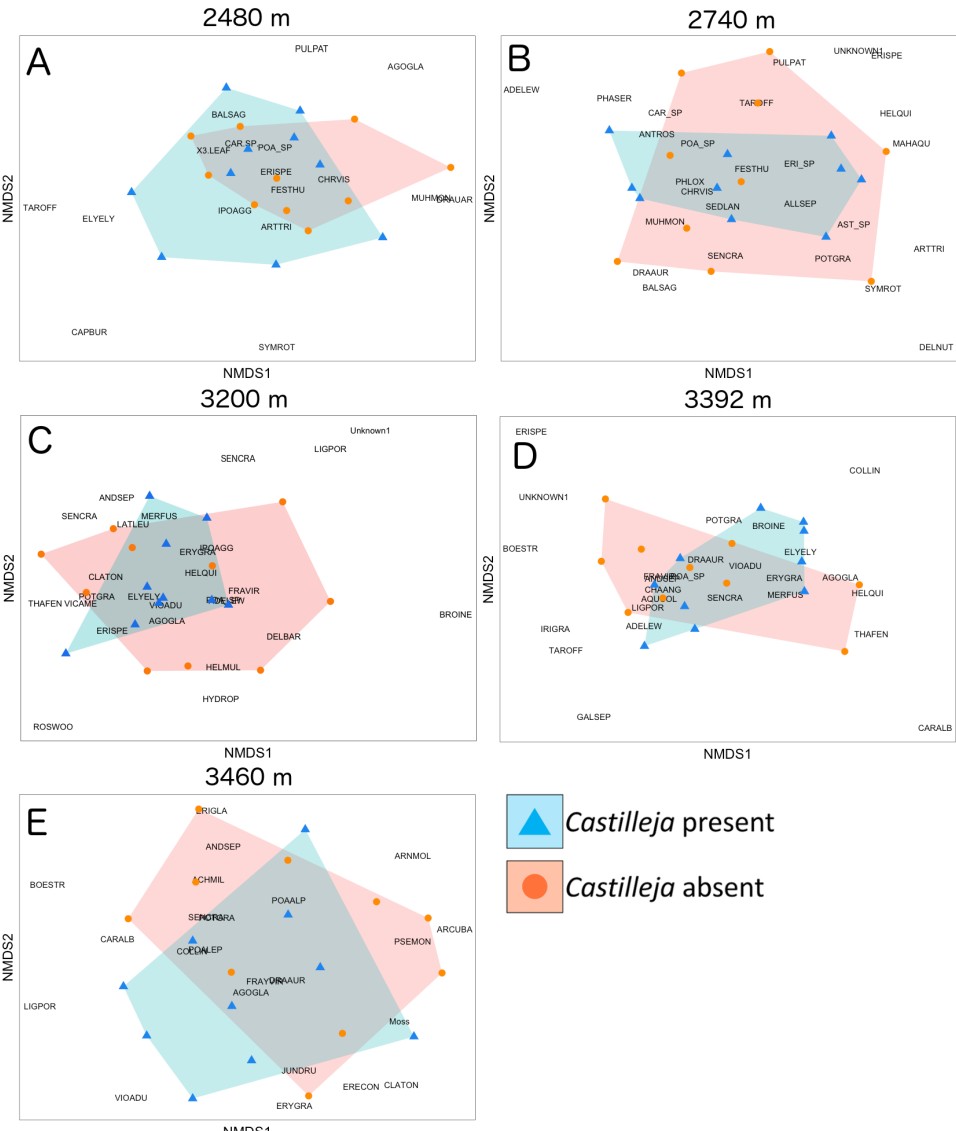

**Figure 2 Non-metric multidimensional scaling of plant community composition based on Bray–Curtis distances with either *Castilleja* present (blue) or absent (orange) across five elevational sites.** (A) 2,480 m, (B) 2,740 m, (C) 3,200 m, (D) 3,392 m, (E) 3,460 m. Plant species identities: ACHMIL, *Achillea millefolium*; ADELEW, *Adenolinum lewisii*; AGOGLA, *Agoseris glauca*; AGO_SP, *Agoseris* sp.; ALLSEP, *Allium* sp.; ANDSEP, *Androsace septentrionalis*; ANTROS, *Antennaria rosea*; ARCUVA, *Arctostaphylos uva-ursi*; ARNMOL, *Arnica mollis*; ARTTRI, *Artemisia tridentate*; BALSAG, *Balsamorhiza sagittata*; BOESTR, *Boechera stricta*; BROINE, *Bromopsis inermis*; CAPBUR, *Capsella bursa-pastoris*; CARALB, *Carex albonigra*; CAR_SP, *Carex* sp.; CHRVIS, *Chrysothamnus viscidiflorus*; CLATON, *Claytonia lanceolata*; DELNUT, *Delphinium nuttallianum*; DRAAUR, *Draba aurea*; ELYELY, *Elymus elymoides*; ERECON, *Eremogone congesta*; ERISPE, *Erigeron speciosus*; ERI_SP, *Erigeron* sp.; ERYGRA, *Erythronium grandiflorum*; FESTHU, *Festuca thurberi*; FRAVIR, *Fragaria virginiana*; GALSEP, *Galium septentrionale*; HELQUI, *Helianthella quinquenervis*; IPOAGG, *Ipomopsis aggregate*; JUNDRU, *Juncus drummondii*; LIGPOR, *Ligusticum porteri*; MAHAQU, *Mahonia repens*; MUHMON, *Muhlenbergia montana*; PHASER, *Phacelia sericea*; POA_SP, *Poa* sp.; PHLOX, *Phlox hoodii*; POTGRA, *Potentilla gracilis*; PSEMON, *Pseudocymopterus montanus*; PULPAT, *Pulsatilla patens*; ROSWOO, *Rosa woodsii*; SEDLAN, *Sedum lanceolatum*; SENCRA, *Senecio crassulus*; SYMROT, *Symphoricarpos rotundifolia*; TAROFF, *Taraxiacum officinale*; THAFEN, *Thalictrum fendleri*; VIOADU, *Viola adunca*.

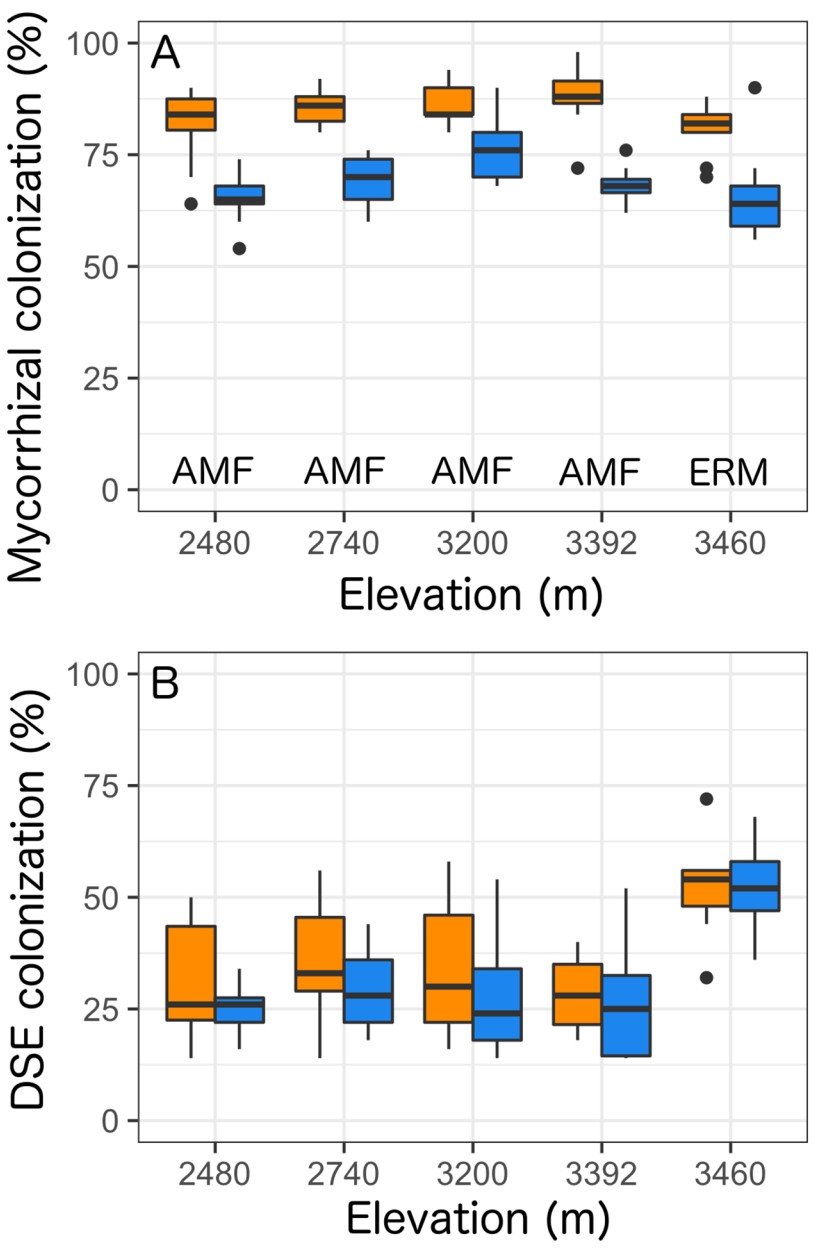

**Figure 3** Colonization rates of (A) mycorrhizal fungi and (B) dark-septate endophytes (DSE) across all five elevational sites with *Castilleja* present (blue) or *Castilleja* absent (orange). Mycorrhizal fungal colonization consists of arbuscular mycorrhizal fungi (AMF—at 2,480, 2,740, 3,200, 3,392, 3,460 m) or ericoid mycorrhizal fungi (ERM—3,460 m). Each boxplot consists of ten plots at each elevation site. The midline represents the median value, with lower and upper hinges corresponding to the 25th and 75th percentiles. The upper and lower whiskers extend from the hinges to highest to the lowest values but no further than 1.5× the inter-quartile range (IQR).

*Chrysothamnus viscidiflorus* (2,740 m), *Viola adunca* (3,200 m), *Ligusticum porter*i (3,392 m), and at our highest (3,460 m) elevation site, *Castilleja* presence was correlated to a reduction in ericoid mycorrhizae colonization within *Arctostaphylos uva-ursi*. We found that climate contexts had no impact on *Castilleja* effects, as elevation was not retained within our best-fit model for mycorrhizal colonization (Table S1). Surprisingly, we found

that *Castilleja* presence was not related to DSE colonization and only elevation was retained within the best fit model predicting DSE colonization. Overall, we found that DSE colonization was highest ($\mu$ = 51.9%) at our high elevation site (3,460 m), and colonization levels were similar at the four other elevations, ranging from 27% to 33% (Table 2; Fig 3).

# DISCUSSION

## *Castilleja* increases plant richness and evenness

Overall, we found the presence of *Castilleja* was associated with an 11% increase in plant richness and a 5% increase in plant evenness. Our results are in line with several studies that have found increased plant richness and evenness with the presence hemiparasites (*Davies et al., 1997*; *Press & Phoenix, 2005*; *Bardgett et al., 2006*; *Westbury et al., 2006*; *Reed, 2012*), although this result can be contingent on hemiparasite identity and the identity of the surrounding plant community (*Press & Phoenix, 2005*; *Westbury & Dunnett, 2007*). Interestingly, we found the effect of *Castilleja* associated with plant richness and evenness did not differ by elevation, suggesting in this system *Castilleja* effects may be consistent despite the shifting biotic and abiotic contexts along the gradient. Plant richness and evenness shifted across the elevational gradient, displaying the characteristic unimodal pattern with highest richness and evenness at middle elevations. Additionally, the species of *Castilleja* present differed by elevation, with *C. angustifolia* present at 2,480 and 2,740 m, *C. miniata* at 3,392 m, and *C. sulphurea* at 3,200, 3,392, and 3,460 m.

## *Castilleja* has inconsistent effects on plant community composition

The presence of *Castilleja* was associated with a significant, but weak, effect on overall plant community composition across the elevational gradient. However, site-level analyses revealed that the effect of *Castilleja* on abundance-weighted plant community composition differed by site. For example, at two of our middle elevation sites (2,740 and 3,200 m), the presence of *Castilleja* was the strongest predictor of plant community composition. The presence of *Castilleja* was correlated with a reduction in aboveground cover of shrub species *Artemisia tridentata* and *S. rotundifolia* at 2,740 m and invasive grass *Bromopsis inermis* at 3,200 m. The reduction of shrub and grass taxa was matched with an increase in low abundant forb taxa: *Adenolinum lewisii*, *P. sericea*, and *Antennaria rosea*. However, it is important to note the nature of our study did not allow us determine which hosts *Castilleja* were parasitizing, how much carbon was being parasitized from hosts, or the long-term dynamics of *Castilleja* in an ecosystem comprised mostly of long-lived perennial plant taxa. In a previous study by *Ducharme & Ehleringer (1996)*, authors found *Castilleja* received ~40% of its carbon from critical host *Artemisia tridentata*. However, *Castilleja* had no effect on the photosynthetic rates or water potentials of *Artemisia* over a 2-year period. Thus, it is unclear how detrimental *Castilleja* is for host production and survival (*Ducharme & Ehleringer, 1996*; but see *Reid, Yan & Fittler, 1994*; *Bowie & Ward, 2004*).

The association between *Castilleja* presence on plant community composition was only apparent when we compared plant community composition using abundance-weighted Bray–Curtis distance metrics. When we compared plant community composition using presence/absence data we found no association of *Castilleja* presence. This suggests that the association of *Castilleja* is primarily driven by abundance changes among dominant plant taxa and less by taxa gains and losses. Taken together, our results align with previous studies demonstrating that few hemiparasites reduce host resources enough to cause mortality, but reduce host growth rates which allow the establishment of subordinate taxa (*Reid, Yan & Fittler, 1994*; *Watson, 2009*).

Although our study is not a direct test of the effect of *Castilleja* on plant community composition, our results align with previous direct, manipulative hemiparasite removal and addition experiments that report a strong negative effect of hemiparasites on dominant taxa abundance with an overall increase in plant diversity, especially with forb taxa (*Pennings & Callaway, 1996*; *Davies et al., 1997*; *Press, 1998*; *Press & Phoenix, 2005*; *Bardgett et al., 2006*; *Těšitel et al., 2015*, *2018*). For instance, sowing of hemiparasite *Rhinanthus* into a *Calamagrostis* invaded grassland reduced *Calamagrostis* cover from ~45% to ~2% within two growing seasons (*Těšitel et al., 2018*). Overall, plant diversity increased to compensate for the reduction in *Calamagrostis* biomass and cover, although authors found that *Rhinanthus* had inconsistent effects on overall plant community composition. Taken together, this suggests that hemiparasite effects on plant community composition are likely driven by the identity of the surrounding plant species, competitive ability of surrounding plant taxa, and abundance levels of surrounding plant taxa (*Press & Phoenix, 2005*; *Watson, 2009*; *Těšitel et al., 2018*). To confirm previous findings, a direct test manipulating both plant community composition and the presence of hemiparasites needs to be performed.

## Site-level spatial heterogeneity was a strong driver of plant community composition

At three of the five sites (2,480, 3,392, 3,460 m), plot pairing was the strongest predictor of plant community composition explaining ~13%, 18%, and 13%, respectively, of the data variation in community composition. At each of these three sites, the presence of *Castilleja* had no effect on plant community composition, suggesting that site-level spatial heterogeneity obscures our ability to detect changes in community composition mediated by hemiparasites. This was surprising because we observed a similar increase in plant richness and plant evenness between plot pairings at all five sites. The overall richness and evenness effects, but limited compositional effects likely reflects the generalist hemiparasite life history of *Castilleja* (*Ducharme & Ehleringer, 1996*; *Adler, 2003*; *Spasojevic & Suding, 2012*). Additionally, the effect of plot pairing was stronger using abundance-mediated measures of community composition (Bray–Curtis distance), compared to presence-absence weighted-metrics (Jaccard's distance). With the exception of our low elevation site, we observed no effect of plot pairing on plant community composition using Jaccard's distance metrics. This suggests that at each site, differences in community composition were driven by abundance differences in taxa plot to plot, and less by taxa turnover among plots.

## Hemiparasitic plants shape abundant plant interactions with mycorrhizal fungi

Overall, we found that the presence of *Castilleja* was associated with a reduction in AMF colonization by 18% and ERM by 17%. However, we found no effect of *Castilleja* presence on DSE colonization within focal taxa roots. The presence of *Castilleja* reduced mycorrhizal colonization similarly across elevation sites even though the overall rate of colonization differed by elevation, suggesting a ubiquitous pattern independent of biotic and abiotic contexts, counter to our expectations. However, our results provide an indirect test of the effect of *Castilleja* on mycorrhizal colonization because we did not confirm if sampled plants were actively parasitized by *Castilleja*. Additionally, we were unable to determine the origin and identity of every single root we quantified for colonization. However, even with the indirect measure and the uncertainty of host identity, we still observed a consistent reduction in mycorrhizal colonization when *Castilleja* was present.

Our results align with previous studies that observed a reduction in mycorrhizal fungal colonization with hemiparasite colonization (*Gehring & Whitham, 1992*; *Davies & Graves, 1998*; reviewed in *Press & Phoenix, 2005*). One potential explanation for the reduction in mycorrhizal colonization is that hemiparasites outcompete mycorrhizal fungi for host carbon resources (*Gehring & Whitham, 1992*; *Davies & Graves, 1998*; reviewed in *Press & Phoenix, 2005*). Because AMF have a higher reliance on host-derived carbon resources relative to DSE and ERM, competition with hemiparasites for host carbon may explain why we observed reduced AMF colonization, but no difference in DSE colonization (*Caldwell, Jumpponen & Trappe, 2000*; *Jumpponen, 2001*; *Usuki & Narisawa, 2007*; *Knapp et al., 2018*). Although at our high elevation site, we observed reduced ERM colonization in *Arctostaphylos uva-ursi* even though ERM produce a suite of carbon-degrading enzymes making them less reliant on plant-derived carbon (*Read, Leake & Perez-Moreno, 2004*; *Smith & Read, 2008*; *Averill, Turner & Finzi, 2014*). Future studies should directly test whether hemiparasites reduce carbon allocation to mycorrhizal fungi and whether the differences in the reliance on host carbon resources will determine the response of fungal colonization to hemiparasite presence.

Hemiparasites also directly associate with mycorrhizal fungi and DSE (*Bouwmeester et al., 2007*; *Li & Guan, 2008*; *Stein et al., 2009*; *Li et al., 2012*). Associations with mycorrhizal fungi and DSE may provide an additional mechanism for hemiparasites to access soil and plant-derived resources (*Li & Guan, 2008*), so it is unclear whether a reduction fungal colonization in dominant plant species would be detrimental to hemiparasites. In a previous field census, we found both AMF and DSE actively colonizing the roots of *C. angustifolia*, *C. miniate*, and *C. sulphurea* across this elevational gradient (J. Henning, 2013, 2015, unpublished data). However, we did not measure fungal colonization rates of *Castilleja* in this study. Our results highlight an interesting pattern, however further study is required to determine if interactions among hemiparasites, mycorrhizal fungi, and plant hosts could cascade to influence plant community composition and ecosystem function.

### Hemiparasite effects on plant diversity, plant community composition, and mycorrhizal colonization are independent of climate contexts

We sought to explore how climate contexts shape the impact of hemiparasites on plant richness, evenness, community composition, and the relationship of dominant plant species with root symbionts. Although we found that the presence of *Castilleja* or elevation were always retained in the best fit models for plant richness, evenness, mycorrhizal colonization, and DSE, we did not observe any significant interactions between the presence of *Castilleja* and elevation for any response variable. This suggests that hemiparasites increase plant richness, increase plant evenness and reduce mycorrhizal fungal colonization within dominant species independent of biotic and abiotic contexts. Our results are counter to several studies, which have found hemiparasite effects are contingent on-site properties like: plant composition; nutrient availability, soil moisture, mycorrhizal fungi present, plant productivity (*Callaway & Pennings, 1998*; *Matthies & Egli, 1999*; *Stein et al., 2009*; *Těšitel et al., 2015*, *2018*). Thus, across a wide array of ecosystems, hemiparasites may be critical for the maintenance of plant diversity and the regulation of competitively dominant plant taxa.

## CONCLUSIONS

Overall, we found that the presence of *Castilleja* was consistently associated with increased richness and evenness within the plant community, while reducing colonization by mycorrhizal fungi across a wide-spread elevational gradient. This suggests that the effect of *Castilleja* on plant and fungal communities is consistent across climate contexts, the species of *Castilleja* present, differences in plant community composition, and differences in underlying overall plant diversity and evenness. Surprisingly, even with the consistent diversity and evenness responses observed within the plant community, the overall effect of *Castilleja* on plant community composition was inconsistent. Although our study was observational by design, our study provides testable hypothesis to explore within future mechanistic experiments exploring the interaction between hemiparasites, mycorrhizal fungi, and host plants, across climatic gradients.

## ACKNOWLEDGEMENTS

We like to thank the staff of the Rocky Mountain Biological Laboratory (RMBL) for their logistical. Additionally, we would like to acknowledge Quentin D. Read for the mental and emotional contributions while establishing and collecting background data from the elevational sites in collaboration with Jeremiah A. Henning. Finally, we would like to thank Petr Blažek and an anonymous reviewer for contributions and helping us substantially improve our manuscript.

### Funding

This work was funded by the Department of Ecology and Evolutionary Biology at the University of Tennessee (Jeremiah A Henning), the RMBL Dr. Lee R. G. Snyder Memorial

Fellowship (Jeremiah A Henning), and the RMBL Fran Hunter Fellowship (Jeremiah A Henning). There was no additional external funding received for this study. The funders had no role in study design, data collection and analysis, decision to publish, or preparation of the manuscript.

### Grant Disclosures
The following grant information was disclosed by the authors:
Department of Ecology and Evolutionary Biology at the University of Tennessee.
RMBL Dr. Lee R. G. Snyder Memorial Fellowship.
RMBL Fran Hunter Fellowship.

### Competing Interests
The authors declare that they have no competing interests.

### Author Contributions
- Michael McKibben conceived and designed the experiments, performed the experiments, analyzed the data, prepared figures and/or tables, authored or reviewed drafts of the paper, approved the final draft.
- Jeremiah A. Henning conceived and designed the experiments, performed the experiments, analyzed the data, contributed reagents/materials/analysis tools, prepared figures and/or tables, authored or reviewed drafts of the paper, approved the final draft.

### Field Study Permissions
The following information was supplied relating to field study approvals (i.e., approving body and any reference numbers):
The sites are located on USDA Forest Service land and are covered under Forest Service Special Use Permit #GUN1120.

### Data Availability
The raw data are provided in the Supplemental Files.

### Supplemental Information
Supplemental information for this article can be found online at http://dx.doi.org/10.7717/peerj.5682#supplemental-information.

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
