# Peer review of "Hemiparasitic plants increase alpine plant richness and evenness but reduce arbuscular mycorrhizal fungal colonization in dominant plant species"

_PeerJ, doi:10.7717/peerj.5682_

## Round 0.1 · original submission · Major Revisions

All three reviewers identify weaknesses in the analysis and interpretation of the manuscript, however, they do provide advice on how to remedy these issues. Please work through the advice of the reviewers in detail.

Reviewer 1 ·

Basic reporting

• good English (only the first sentence in the conlusions need rephrasing to be easier to follow)
• intro and literature good, but could be improved
• normal structure
• figures relevant, but Fig. 2 with different elevation for the highest site and missing description of boxplot charasteristics (median, first and third quartile) and whiskers (2% and 98% percentile ???)
• raw data provided

Experimental design

• primary research reported
• questions well defined
• knowledge gap is not properly identified
• rigorous investigation performed

Validity of the findings

• impact of findings not well assessed, rather descriptive and vague
• data is robust but interpretation is not well balanced, need to take into account the limitations of experimental design
• last paragraph commenting effects of global changes is not relevant to the study
• conclusions are well supported

Additional comments

This study addresses an important issue whether hemiparasitic plants (from genus Castilleja) consistently increase diversity of plant communities across an elevational gradient and if their impact can be connected to disrupted root mycorrhizal colonisations of the plant community dominants. Especially the second part could represent substantial contribution to the respective field.
Research questions are properly defined, data collected in the field survey correctly analysed and presented using suitable diagrams and tables. However, the interpretation of the results lacks the view of plant ecologist and discussion of the findings is sometimes rather vague (see specific points below).
Biology of hemiparasitic Castilleja plants was not well introduced and also acknowledged when planning field sampling design for this study. Comparison of adjacent plots with and without hemiparasites (from visual estimates of Castilleja presence in the year of survey) does not enable to make strong inference about effect of this functional group on plant community functioning. This would need experimental manipulation (either introduction or removal of Castilleja) and more years of plot monitoring.

Major study caveats:

1) Samples of roots selected for evaluation of mycorrhizal status (lines 123 – 140) were not checked for the connection of haustoria from Castilleja plants. Thus, this study does not bring a direct test whether hemiparasites decrease colonisation of host roots by mycorrhiza.

2) According to methodology (lines 118 – 119) the dominant plant species for evaluation of the mycorrhizal status should be the species with ‘the highest relative abundance across 20 plots recorded at each site’. However, having the raw plant community data in hand this rule was respected by the authors only at one site – i.e. with the lowest elevation (2480m, Balsamorhiza sagittata) but not in other 4 sites. Correctly should be used following species: 2740m (Phlox hoodii with 37% relative abundance instead of Chrysothamnus viscidiflorus with 22%); 3200m (Helianthella quinquenervis with 11% instead of Viola adunca with 9%); 3392 m (Potentilla gracilis with 16% instead of Ligusticum porter with 15%) and 3460m (Poa alpina with 17% instead of Arctostaphylos uva-ursi with 7%).

3) Missing raw mycorrhizal data in the Excel sheet (showing only 98, missing 2 from 3200m site).


Minor points to be solved:

Lines 44 – 47: Hemiparasitic plants can reduce the abundance of dominant plant species, but this feature is rather limited to ‘root hemiparasites’ (not stem).

Lines 52 – 54: It needs to be emphasized that hemiparasites from definition (hemi) are dependent on host in the case of water and dissolved mineral nutrients (N, P, K, Ca, Mg etc.) but carbon received from parasitism is of minor importance for them. Hemiparasites are green plants and do photosynthesis and carbon assimilation on their own (in contrast to holoparasites).

Lines 54 – 55: Indeed ‘hemiparasites may compete with host plant plants for soil resources through absorptive roots’ but this is not ‘additionally’ to above mentioned ‘haustoria’ (line 53). This is connected to the same root structure.

Lines 62 – 63: ‘Hemiparasite colonization reduces carbon allocation to fungal symbionts’ may be rewritten to the following: Attack by hemiparasites reduces host’s carbon allocation to fungal symbionts.

Lines 80 – 84: Please provide some details on biology of Castilleja species under investigation: whether they are biennial / perennial, clonal; which hosts do they prefer.

Line 83: Castilleja rhexiifolia was present according to Table 1 only at two sites (3200 and 3392 m).

Lines 116 – 117: ‘Percent coverage was estimated to the nearest 1% for species >20% and the nearest 5% for 117 species <20% coverage.’ Please first use <20% and then >20%.

Line 174: ‘The presence of Castilleja increased plant richness by 11% (F1,90=33.64, p > 0.0001), increased Shannon’s diversity by 9% (F1,90 = 13.82, p > 0.0003)’. Please replace it with p < 0.0001 and p < 0.0003, respectively.

Line 264: Please correct occurrence of C. rhexiifolia according to Table 1 (i.e. 3200 and 3392 m).

Lines 267 – 275: Out of focus of the study, remove it.

Lines 278 – 281: Please rephrase it with a couple of simple sentences.

Line 306: Correct to Ecology Letters (with a capital letter for the second name).

Lines 309, 318, 336, 354: Please put ‘and’ in front of the last author names.

Line 328, 394, 397 : Please give journal titles in italics.

Line 412: Please correct the legend of Fig. 1 according to diagram. There is no Simpson’s index.

Line 432: Please correct TAROFF = Tarax(i)acum officinale.

Line 444: Calculation of ‘Simpson’s diversity’ was neither shown in the Methods nor reported in the Results.

Figure 1 and 3: Please indicate the values which are shown by boxplot (median, first and third quartile) and especially whiskers (???).

Figure 2: in the legend please correct BALSAG = Balsamorhiza sagittata (e); TAROFF = Tarax(i)acum officinale; in the diagram use uniformly the same elevation for the highest site as anywhere else in the text (i.e. 3460 m).

Figure 3: Please say in the legend what the different colours (blue and orange) show (Castilleja presence and absence).

Table 2: Please change the ‘>’ signs to ‘<’ for all P values (last column).

Reviewer 2 ·

Basic reporting

There are quite a few minor errors in spelling, grammar, and punctuation throughout. The literature cited does not always back up or refer to the points made in the article, and this was particularly common in the introduction. The sections on fungi and climate definitely need to be improved as they do not make the best use of the literature. The article is structured well but the figure and table legends need to be clearer so that the reader can interpret them without referring to the main text.

I applaud the authors for making their raw data available along with their R-code. This really useful.

I have made specific line by line comments in the general comments section below, which will hopefully assist the authors.

Experimental design

The design appears to be mostly sound, although a figure of the actual layout would be really helpful (randomised but adjacent sampling could be more clearly expressed in a figure). The root sampling is not convincing from the point of view of only collecting the target species. I think the authors need to acknowledge that they cannot be certain that the roots examined come from the target plant alone.

Research questions were well defined, relvant, and meaningful.

Technical and ethical standards are high, but choice of statistical analyses is incorrect in places.

Replication of the methods will be possible once the randomised adjactent sampling is better explained.

Validity of the findings

The main issue with the article is the use of simple linear models to assess data that includes binary variables and species counts. GLM or mixed model approaches are needed, and it would be useful to note whether the distribution and variance of the data was examined. The use of ordination techniques could also be improved and further explored by exploring the correct number of dimensions for NMDS and applying MRPP or PerMANOVA approaches. Once the authors have revised their analyses to use the correct techniques then it will be easier to assess the validity of the findings.

Additional comments

In general I think there is a really good paper in here that just needs some more thought and careful re-examination of the data with the correct statistical techniques. The study system is really interesting and the results of the study will be of interest to readers in a few different fields of ecology.

Specific comments
Introduction, ln 48: abundance, not abundant
Ln 53: haustoria, not haustroria
Ln 56: remove comma
Lns 54-58: This sentence is a bit unclear and needs to be more specific. Is the suggestion here that the hemiparasites may form mycorrhizas with AMF that compete with the AMF of the host plant for resources? Or do you mean that the hemiparasites associate with the same AMF as the host plant (e.g. via a common mycorrhizal network) and reduce the quantity of nutrients being transferred to the primary host? How would DSEs help a hemiparasite to compete for soil resources? It would be appropriate to provide a reference for this, although I’m not aware of any evidence for DSEs providing competitive benefits to hemiparasites (Li and Guan record colonisation in their 2007 paper but I don’t they studied impacts on competition).
Lns 59-60: The Joshi et al 2000 paper does not measure belowground biomass or even discuss fungi. Here you are speculating that AMF and DSE may have an additional effect beyond what has been investigated in the literature, but this sentence reads as though it has been previously explored and is an accepted idea.
Lns 60-62: That is a strong statement to make about DSEs. The Mandyam & Jumpponen paper you cite makes the point that responses to DSEs are hugely variable. I don’t recall there being any evidence of nutrient uptake by DSEs, so you might want to clarify whether you mean growth benefits or something else.
Ln 62-63: Should be (Stewart and Press, 1990). This reference does not make that claim. Your statement is one possible hypothesis that can be drawn from the paper, but they certainly don’t test it or even mention it. It could also be that hemiparasite colonisation has no effect on the carbon allocated to fungal symbionts, or perhaps the plant allocates more carbon to the symbionts in an effort to increase its access to nutrients.
Ln 63-66: This is better. It would be useful to note that this has been shown in controlled pot experiments.
Ln 67: The Stewart and Press reference does not mention this concept.
Ln 69-70: Choice of references is a bit odd again, but maybe it is just the sentence structure. If you start out by noting that climate contexts shape fungal and plant community structure and function (there are lots of more recent papers on the impacts of climate on plant and fungal communities), you can then say that hemiparasites are likely to interact with these effects.
Ln 70-71: So do beneficial mutualistic interactions. I think you could strengthen this section (lns 69-79) through better choice of references and more careful wording. Section starting Lns 73-79 is good, but 69-73 is weak (including a sentence that cites the same two papers twice).

Ln 91: Point 2) does not flow properly in the context of a list beginning with “We hypothesised that:” You could change it to 2) the presence of Castilleja would alter community composition.
Methods, ln 104: You have a superscript 1 after cm.
Ln 113-114: Can you put in a figure to illustrate this? I’m curious about how this was done in practice as 20 plots per elevation is easy to understand, but randomly selected adjacent plots with and without Castilleja is not so clear. There is the potential for spatial autocorrelation in your sampling so it would be useful to give the reader more information.
Lns 123-140: So this is just a bulk root sample, you didn’t try to identify whether the roots came from one or multiple host plants / host plant species?
Ln 144: plant species richness
Ln 145-149: This is problematic. You do not indicate having examined the distribution or variance of the data, and one of your variables is binary (present/absence). Really you should be using a glm (Generalised linear model) or LME (linear mixed effect model) function not a simple linear model. This will allow you to account for a binomial error structure by including a logit link function or a poisson error structure using a log link function (plant species richness is count data). You should also include some species accumulation curves (could be supplemental) for your plant community data to show whether you reached saturation or not.
Ln 158-164: Using NMDS you could include all of the elevations together to see if they differ from each other. You would also want to use an MRPP or PerMANOVA analysis to examine whether + and – Castilleja groups are considered significantly different or not. The problem with using R to do NMDS is that you are telling it to look at two axes without knowing how many axes best explain the data. I’m not sure if there is a workaround because I do not use R for doing ordinations. It may be that a 3D fit is best, which could change things substantially.
Ln 165-170: Same problem as lns 145-149 above, you cannot use a linear model to examine a binary predictor in this way. You also need to adjust the language to note that the roots are not necessarily just from the dominant host.
Results will all need to be re-assessed following analysis with correct models. Subheadings should reflect the hypotheses being tested, rather than the interpretation of the results.
Discussion will also need to be reassessed following corrected analyses.
ln 216: low abundance, not low abundant. Should be a full stop after diversity, a comma after however, and insert “the presence of these plants” or something similar.
Ln 234-236: From the way the methods are written you cannot be sure that those roots all came from the dominant host, so be careful with the statements about what you can show here.
References ln 398: the link is broken
Figure 1. Define what you mean by plant community. Legend details are a bit sparse. The reader should have all of the information they need in the figure legend.
Figure 2. How many ordination axes did you examine? If you have three significant axes then those communities could be separated in ordination space.

·

Basic reporting

Meets the standards of PeerJ

Experimental design

Generally appropriate experimental design and data analysis, but see my suggestions for improvement below.

Validity of the findings

The data and analyses are robust, but the interpretation is not fully supported by the results, especially since the causality is unclear in observational study. See detailed comments below.

Additional comments

Review on manuscript 28743
Hemiparasitic plants increase alpine plant diversity but reduce arbuscular mycorrhizal fungal colonization in dominant plant species
Submitted to PeerJ

Recommendation: Major revision

The study reported in this manuscript aimed to explore the effect of a hemiparasitic plant genus on plant species diversity, composition and root fungal colonization in an altitudinal gradient. The research questions are well supported by the introduction, the sampling was well designed and data processed by appropriate methods. The authors are native speakers, unlike me, and the manuscript was easily intelligible to me. The study is generally good to be published in PeerJ, but it needs a revision of certain points. Especially, I am very doubtful about the interpretation of the observed patterns as causal relationships, and I find the selection of diversity measures misleading. In addition, I have many minor comments, which need the authors’ attention.

Comments to the authors:
Throughout the whole manuscript, you state that “presence of Castilleja impacted / increased / reduced... something”. However, this is an observational study and you have no evidence that the observed statistical relationships are caused by the parasitic behaviour of Castilleja. For instance, a related hemiparasitic plant Rhinanthus minor is also known to reduce community biomass and sporadically increase plant richness, but at the same time it fails to establish in more productive and less diverse communities, and it is impossible to disentangle these two processes without a manipulative experiment. Your study provides valuable data, but the observed patterns must be interpreted appropriately, e.g. “presence of Castilleja was associated with higher species richness...”.
Hejcman, M., Schellberg, J., Pavlů, V., 2011. Competitive ability of Rhinanthus minor L. in relation to productivity in the Rengen Grassland Experiment. Plant Soil Environ. 57, 45–51.
van Hulst, R., Shipley, B., Thériault, A., 1987. Why is Rhinanthus minor (Scrophulariaceae) such a good invader? Can. J. Bot. 65, 2373–2379.
Westbury, D.B., Davies, A., Woodcock, B.A., Dunnett, N.P., 2006. Seeds of change: the value of using Rhinanthus minor in grassland restoration. J. Veg. Sci. 17, 435–446.

Lines 144 – 145, Table 2: I do not like using all of these measures: richness, evenness, Shannon’s diversity and Simpson’s diversity. Richness and evenness are independent of each other, but both diversity measures are just composed of richness and evenness in various proportion. Therefore the p-values for the diversity measures are in between the ones of richness and evenness, and Simpson is less significant than Shannon. Statements like “Castilleja increased plant species richness and diversity” (e.g. lines 31 – 32, 209 – 210) seem to present two effects, but it is actually just one. Moreover, you present Simpson only in the table, not in the text.

L. 21 – 22, 44 – 51: Some hemiparasitic plants do not always provide the desirable effect.
Westbury, D.B., Dunnett, N.P., 2007. The impact of Rhinanthus minor in newly established meadows on a productive site. Appl. Veg. Sci. 10, 121–129.
L. 84, 218: Is there any information available about the level of host specificity of Castilleja species?
L. 104 – 105: I guess that you used mm/yr (not cm/yr). What are actually these values? Why do they not correspond to “MAP (mm)” in Table 1? The same for temperature.
L. 113 – 114: Could you honestly describe, how did you select the experimental plots? Real random selection would hardly result in such a nice pair design. How did you select plots with the parasite and how did you select the adjacent control plots?
L. 119 – 121: Statement “Next, we collected a single root core...” fits better to the next section.
L. 124 – 126: The calculation of number of samples in brackets repeats previous paragraph, and moreover is not completely true. I wondered why there are only 88 residual df in Table 1 for AMF and DSE colonization when you state in methods you have 100 soil cores. I found only in supplementary material that there were 9 – 11 data points in each site*treatment. You should state in methods something like “about 10 plots” and “98 total soil cores” and mention the cause of missing data.
L. 145: Typo in the formula, should be “H= - Σpiln(pi)” (lowercase subscript i, not “H= - ΣpIln(pi)”)
L. 147 – 148: Statement “... by constructing linear models with elevation (m),...” clearly implies that numeric variable Elevation was used in general linear model, which is not true. Categorical variable Site was in fact used in analysis of variance (or general linear model, as you wish) and the statement should be rephrased appropriately.
L. 151 – 155: These lines can be omitted. The first sentence describes well-known principles of ANOVA which is redundant here, the last sentence fits more to discussion.
L. 158 – 162: nMDS does not perform a statistical “test”, you should better use a synonym which is not a statistical term (e.g. “compare”?). Moreover, I think that statements “Castilleja influenced plant community composition” and “certain plant species were impacted more or less by Castilleja” mean the same and the latter sentence can be omitted.
L. 173: The headings within Results and Discussion sections sound very weird to me. You should not say the result in the heading, especially in the Results section (I could accept it in Discussion).
L. 180 – 183, 191 – 193: Do not interpret the results in the Results section, you can omit these lines.
L. 189 – 190. You cannot interpret p = 0.045 if you have 5 related tests (think about multiple testing and false discovery). Also the 3 out of 5 trends in the same direction mean nothing, it cannot be closer to the null hypothesis 2.5 out of 5.
L. 201 – 204: Remove brackets from DF.
L. 211 – 221: Do not refer to “our first/second... hypothesis”, the reader does not remember what they were. Consider rephrasing (maybe “contrary to our expectations” instead of “leading us to reject our second/fourth hypothesis”?)
L. 261 – 262: Statement “Additionally, the effects... did not vary by species...” sounds to me as if you were introducing an additional result, but it is just an alternative interpretation of the previous result with unknown cause. Consider rephrasing. For instance, for several reasons mentioned above, you can rephrase l. 218 – 221 “Surprisingly, we found that positive hemiparasite association with plant richness was independent of site differences (possibly related to climate, community composition or Castilleja species present).”
L. 280: “reduced”
L. 281: “are highlighting” or “highlighted”
L. 435: Really also AMF in 3460m?
L. 441: Abbreviation of fungi are missing (AMF, DSE and especially ERM, which is mentioned here first time)
Figure 2: Disproportionate labels. If the figure is printed in one column, the species labels are too small. If it is printed in two columns, the altitude and treatment labels are too large.
Table 1: A row with summary of Castilleja abundance would be interesting. In addition, the row “Castilleja sp. present” does not fit to the raw plant community data. I see in the raw data: 2480 and 2740 CASANG (what is this species?), 3200 CASSUL, 3392 CASSUL and CASMIN, 3460 CASSUL.
All “>” and “<” signs are in the wrong direction. I guess that e.g. P<0.0001 or that you estimated coverage to the nearest 5% for species >20% coverage.
The figure and table captions on lines 410 – 447 differ from the captions on the following pages. Some captions do not fit the content of the figures and some captions are obviously incomplete.

Petr Blažek

České Budějovice, 30 June 2018

---

## Round 0.2 · accepted · Accept

There are a few very minor corrections to make but these do not require a further referee review. Please incorporate the minor corrections made by the referees while in Production.

# Reviewer 1 ·

Basic reporting

OK

Experimental design

OK

Validity of the findings

OK

Additional comments

I am satisfied with responses, all corrections and changes were made by the authors with respect and the manuscript now reads well. Only correct species name to 'Rhinanthus' on line no. 335 and ensure that species abbreviations in Fig.2 do not overlap.

·

Basic reporting

no comment

Experimental design

no comment

Validity of the findings

no comment

Additional comments

Review on on a revised manuscript 28743
Hemiparasitic plants increase alpine plant richness and eveness but reduce arbuscular mycorrhizal fungal colonization in dominant plant species
Submitted to PeerJ

The manuscript was substantially improved and all major comments were taken into account by the authors. The data analysis was improved to match the structure of the data and to provide statistical tests for the research questions. Also the results are interpreted carefully, taking into account the observational type of sampling. The manuscript can be accepted for publication, I just noted two very minor points which should be solved before final publication (I do not need to see the revised version).

L. 123 - 128: The grammar of this part is wrong and “the presence of Castilleja” is repetitive. Either just move “1)” after “how” (“we measured how 1) the presence of Castilleja…”), or preferably rephrase to: “Here, we measured how the presence of Castilleja 1) was related to plant richness and evenness of the surrounding plant community, 2) would alter community composition, and 3) would reduce dominant plant associations…”

L. 272: presence should not be italic

Petr Blažek

České Budějovice, 28 August 2018